# Cell-Type Specific Regulation of Cholesterogenesis by CYP46A1 Re-Expression in zQ175 HD Mouse Striatum

**DOI:** 10.3390/ijms241311001

**Published:** 2023-07-02

**Authors:** Katleen Pinchaud, Chloé Masson, Baptiste Dayre, Coline Mounier, Jean-François Gilles, Peter Vanhoutte, Jocelyne Caboche, Sandrine Betuing

**Affiliations:** 1Neuroscience Paris Seine, Institut de Biologie Paris-Seine (IBPS), CNRS UMR 8246/INSERM U1130, Sorbonne Université, 75005 Paris, France; katleen.pinchaud@sorbonne-universite.fr (K.P.); chl.masson.cm@gmail.com (C.M.); baptiste.dayre@gmail.com (B.D.); coline.mounier@sorbonne-universite.fr (C.M.); peter.vanhoutte@sorbonne-universite.fr (P.V.); jocelyne.caboche@sorbonne-universite.fr (J.C.); 2Imaging Facility, Institut de Biologie Paris-Seine (IBPS), Sorbonne Université, 75005 Paris, France; jean-francois.gilles@sorbonne-universite.fr

**Keywords:** Huntington’s disease, striatum, CYP46A1, cholesterol, gene regulation, *Srebp2*, *Hmgcr*, *Dhrc24*, *ApoE*, fluorescence in situ hybridization coupled with immunostaining

## Abstract

Cholesterol metabolism dysregulation is associated with several neurological disorders. In Huntington’s disease (HD), several enzymes involved in cholesterol metabolism are downregulated, among which the neuronal cholesterol 24-hydroxylase, CYP46A1, is of particular interest. The restoration of CYP46A1 expression in striatal neurons of HD mouse models is beneficial for motor behavior, cholesterol metabolism, transcriptomic activity, and alleviates neuropathological hallmarks induced by mHTT. Among the genes regulated after CYP46A1 restoration, those involved in cholesterol synthesis and efflux may explain the positive effect of CYP46A1 on cholesterol precursor metabolites. Since cholesterol homeostasis results from a fine-tuning between neurons and astrocytes, we quantified the distribution of key genes regulating cholesterol metabolism and efflux in astrocytes and neurons using in situ hybridization coupled with S100β and NeuN immunostaining, respectively. Neuronal expression of CYP46A1 in the striatum of HD zQ175 mice increased key cholesterol synthesis driver genes (*Hmgcr*, *Dhcr24*), specifically in neurons. This effect was associated with an increase of the *srebp2* transcription factor gene that regulates most of the genes encoding for cholesterol enzymes. However, the cholesterol efflux gene, *ApoE*, was specifically upregulated in astrocytes by CYP46A1, probably though a paracrine effect. In summary, the neuronal expression of CYP46A1 has a dual and specific effect on neurons and astrocytes, regulating cholesterol metabolism. The neuronal restoration of CYP46A1 in HD paves the way for future strategies to compensate for mHTT toxicity.

## 1. Introduction

Huntington’s disease (HD) is a neurodegenerative disorder with autosomal dominant inheritance, onset in young adults, and presents a combination of neuropsychiatric, motor, and cognitive symptoms [1]. The disease is caused by an abnormal expansion of CAG trinucleotide repeats in the gene encoding the huntingtin protein (HTT), resulting in a poly-glutamine repeat in the N-terminal region of the mutated huntingtin protein (mHTT) [2]. The toxic gain of functions of mHTT and the loss of function of wild-type huntingtin result in a cascade of events that leads to a progressive degeneration of medium-sized spiny neurons (MSNs) in the striatum [3], which then extends to other brain regions such as the cerebral cortex, the hypothalamus, and the cerebellum. Multiple cellular and molecular dysfunctions have been well described, including transcriptional gene dysregulation, alteration of energy metabolism, synaptic transmission, BDNF synthesis and transport, TrκB receptor trafficking, clearance of unfolded proteins, and alteration of cholesterol homeostasis [4].

Despite these cardinal discoveries about HD pathogenesis, a significant challenge remains to identify HD modification strategies that could be used to slow disease progression. Current therapeutic approaches focus on: mHTT lowering; inhibition of mHTT aggregation and modulators of key pathways involved in HD pathogenesis such as excitotoxicity, proteostasis, and mitochondrial dysfunction [5,6]; and, more recently, cholesterol metabolism dysregulation [7,8]. 

The maintenance of cholesterol homeostasis is a relevant aspect of the central nervous system (CNS) and its functions, including brain development, myelination, and neuronal signaling and survival. Since peripheral cholesterol cannot cross the blood–brain barrier, brain cholesterol is primarily synthesized locally by astrocytes, while the major pathway for cholesterol catabolism is achieved in neurons by the brain-specific cholesterol 24-hydroxylase enzyme (CYP46A1), leading to the conversion of cholesterol into 24 (S)-hydroxycholesterol (24S-OHC) [9]. Blood cholesterol levels are reduced in manifest HD patients [10,11], and levels of sterols upstream from cholesterol, including lanosterol, lathosterol, and 7-dehydrocholesterol, are markedly decreased within the striatum of HD mice [12,13,14,15,16,17]. Additionally, 24S-OHC levels are reduced in the plasma of HD patients, paralleling caudate nucleus atrophy [11]. The mRNA level of cholesterol biosynthetic genes are reduced in HD cell lines, fibroblasts, and postmortem striatal and cortical tissues from HD patients and in the striatum of several mouse models of HD [18,19,20,21]. In addition, we made the original observation that expression levels of CYP46A1 are strongly reduced in the putamen of HD patients and striatum of both transgenic (R6/2) and knock-in (zQ175) HD mouse models [16,17]. Restoring striatal CYP46A1 levels in neurons by gene therapy using adeno-associated virus (AAVrh10) was neuroprotective in these two HD mouse models, and this approach directly reinstates the whole cholesterol metabolism pathway, including the production of sterols (lanosterol and desmosterol) and 24S-OHC [16,17]. Strikingly, this approach ameliorated the clearance of mHTT aggregates and, most importantly, normalized the striatal transcriptome, especially for genes implicated in synaptic transmission and proteasome activity [17]. 

To better understand how the normalization of CYP46A1 expression reactivates cholesterogenesis, it remains to be determined whether CYP46A1 regulation of cholesterol biosynthetic genes occurs in astrocytes and/or MSNs, the main neuronal population of the striatum. Two non-exclusive hypotheses can be proposed: (i) neuronal expression of CYP46A1 induces a paracrine effect on astrocyte cholesterogenesis through the secretion of 24S-OHC product, which is an activator of nuclear Liver X Receptor (LXR) or, (ii) neuronal expression of CYP46A1 could act locally in neurons to activate cholesterogenesis [22]. To address this question, we took advantages of fluorescent in situ hybridization (FISH) coupled with immunostaining to quantify key cholesterol genes in astrocytes and MSNs of HD mice. We found that the neuronal expression of CYP46A1 in HD zQ175 mice has a local effect on MSNs for the two cholesterol gene drivers, i.e., *Hmgcr* and *Dhcr24*, and a paracrine effect on astrocytes for *Apoe* cholesterol efflux gene. On the other hand, the sterol regulatory element binding protein 2 (SREBP2) transcription factor was regulated in both MSNs and astrocytes after CYP46A1 expression in MSNs. Altogether, these results suggest a bimodal effect of CYP46A1: one in MSNs to reinstate cholesterol biosynthesis and one in astrocytes to favor cholesterol efflux. The beneficial effect of CYP46A1 in HD reflects the dynamic balance between cholesterol synthesis, uptake, and export, all integrated into a dialogue between MSNs and astrocytes. 

## 2. Results

### 2.1. Validation of Virus-Mediated Expression of GFP and CYP46A1-HA in Dorsal Striatum and Validation of FISH Analysis Coupled with Cell-Specific Immunolabelling

#### 2.1.1. Expression of CYP46A1 in the Striatum of HD Mice

In order to study the regulation of genes involved in cholesterol metabolism in HD mice after virus-driven CYP46A1 expression, bilateral stereotaxic injection of AAVrh10-GFP (control) and AAVrh10-CYP46A1-HA (a human version of CYP46A1 with a haemagglutinin (HA) tag) was performed in the striatum of four-month-old mice. Five months later, FISH was performed to quantify gene expression in neurons (NeuN immunolabelling) and astrocytes (S100β immunolabelling) (Figure 1A). The zQ175 HD mouse model did not show lethal phenotypes, and stereotaxic injections (AAVrh10-GFP and AAVrh10-CYP46A1-HA) did not induce toxicity during the timeline of the experiments [17]. We first examined the cell transduction of these two AAVs. As shown in Figure 1B, GFP and CYP46A1-HA expression were strongly expressed in the dorsal striatum five months after stereotaxic injection, as previously described [16,17]. Importantly, double-labelling of HA and a neuronal marker (NeuN) showed that the majority of striatal cells expressing CYP46A1-HA were neurons (Figure 1A, right panels). Therefore, the experimental setup allowed for a long-lasting neuronal expression of CYP46A1 in the striatum of zQ175 mice. 

#### 2.1.2. Imaging Tools to Quantify mRNA Signals after FISH Coupled with IHC

We next sought to visualize specific mRNAs related to cholesterol metabolism in striatal neurons and astrocytes to analyze the effect of CYP46A1-mediated gene regulation in each cell population in HD mice. FISH was performed on 30-µm-thick mouse brain sections using specific fluorophore-coupled RNAscope^®^ probes against *Hmgcr*, *Dhcr24*, *Srebp2*, and *ApoE*. Since each FISH signal gave rise to different patterns, with either sparse or densely packed puncta, a 2D projection (maximum intensity) followed by intensity thresholding (Otsu method) allowed us to measure the number of dots in each nucleus (identified manually by ROIs) with the immunostaining signal (NeuN immunolabelling for neurons and S100β immunolabelling for astrocytes). For *Srebp2* and *Dhcr24,* we used the machine-learning Advanced Weka Segmentation plugin on 2D projection to perform image segmentation based on pixel classification and separate all dots. Then, a threshold and a particle analysis allow us to determine the number of dots in each DAPI-NeuN (Figure 2A) or DAPI-S100β (Figure 2B) labelling. For *ApoE*, which retrieved very dense signals, we used macros in ImageJ to count the dot signal encapsulated in the nucleus in 3D. Due to the heterogeneity of the DAPI and immunostaining signals, we used the deep learning-based plugin Stardist [23] and fine-tuned the training of the model with a small amount of our images for a better segmentation result. Then, cells were manually selected for counting the dot signals in neurons (Figure 2C) or astrocytes (Figure 2D) after a difference of Gaussian filtering and then a 3D Dot Segmentation [24]. Counting was performed with a 3D ROI Manager plugin from the 3D ImageJ Suite [25]. 

### 2.2. Analysis of Hmgcr and Dhcr24, Two Key Cholesterol Synthesis Genes, in Neurons and Astrocytes after CYP46A1-HA Expression in HD Mice

The effect of CYP46A1 in the cholesterogenic pathway was then studied on two critical genes coding the rate-limiting enzyme HMGCR and a key node enzyme, DHCR24, which ensures the link between the Kandutch–Russel and Bloch pathways [7]. *Hmgcr* mRNA FISH signals were detected as discrete dots in the nucleus (Figure 3A). Quantification of *Hmgcr* puncta in neurons (NeuN staining) and astrocytes (S100β staining) did not show any difference between WT-GFP and HD-GFP mice in both cell populations. These results confirm previous studies showing no major dysregulation of *Hmgcr* mRNA expression in the striatum of zQ175 at 12 months [17]. However, CYP46A1 re-expression after AAVrh10-CYP46A1-HA transduction induced a significant increase of *Hmgcr* expression in neurons (Figure 3A,B) but not in astrocytes (Figure 3D,E). MSNs, which comprise around 90% of the neurons in the striatum, belong to two anatomically and functionally distinct populations that exert opposite roles in the selection of motor plans. MSNs of the direct pathway (D1 MSNs) express dopaminergic receptor type 1 (DRD1), while those of the indirect pathway (D2 MSNs) express dopaminergic receptor type 2 (DRD2) [26]. D1 and D2 MSNs exhibit distinctive structural and functional properties in zQ175 mice [27,28,29]. Hence, we sought to analyze if the CYP46A1-mediated increase of *Hmgcr* was more prominent in one of the two MSN populations. We took advantage of the multiplex ability of RNAscope^®^, to analyze the expression of *Hmgcr* in both D1 and D2 MSN populations of HD mice injected with AAVrh10-CYP46A1-HA. As expected, *Drd1* and *Drd2* dots were perfectly segregated in dorsal striatum sections from the HD mice (Figure 3C). In HD-CYP46A1 mice, *Hmgcr* mRNA levels were equally distributed in both D1 and D2 MSN populations (53% in D1 MSNs and 46.9% in D2 MSNs and) (Figure 3C). 

With regard to *Dhcr24* mRNA levels, we did not detect differences in either neurons or astrocytes from WT-GFP and HD-GFP mice. (Figure 4A–D), while CYP46A1 re-expression induced an increase in *Dhcr24* levels in neurons (Figure 4B) but not astrocytes (Figure 4D). 

Altogether, these results show that neuronal CYP46A1 expression has a direct effect on cholesterogenic enzymes within the two MSN populations and no action on astrocytes.

### 2.3. Regulation of Srebp2 Transcription Factor Gene in Neurons and Astrocytes after CYP46A1-HA Expression in HD Mice

Cholesterol biosynthesis is regulated by the transcription factor SREBP2, which activates the expression of most cholesterol biosynthesis genes [30]. The nuclear level and activity of the N-terminal active fragment of SREBP2 are reduced in HD cellular models and mouse brains, and *Srebp2* gene therapy in striatal astrocytes alleviates HD phenotypes in R6/2 mice [31]. We previously showed that the *Srebp2* mRNA level is significantly decreased in zQ175 mice and that CYP46A1 expression tended to restore the proportion of nuclear SREBP2 activity in the striatum of zQ175 [17]. We quantified *Srebp2* mRNA in MSNs and astrocytes to better characterize CYP46A1’s beneficial effect on cholesterogenesis (Figure 5). *Srebp2* mRNA quantification did not show any differences between WT-GFP and HD-GFP groups in either neuronal cells (Figure 5A,B) or astrocytes (Figure 5D,E); however, CYP46A1 significantly increased *Srebp2* mRNA expression in neurons and astrocytes in the HD striatum (Figure 5B,E). The level of *Srebp2* mRNA in HD-CYP46A1 was equally distributed in both D1 and D2 MSNs of HD mice (50.31% and 49.69%, respectively) (Figure 5C). Overall, neuronal CYP46A1 expression in HD mice plays a direct role in the regulation of *Srebp2* gene expression in neurons, with a paracrine effect on astrocytes. 

### 2.4. Neuronal CYP46A1 Expression Effect on ApoE Cholesterol Efflux Gene

APOE protein is mainly expressed by glial cells including astrocytes, microglia, and oligodendrocytes, and is involved in the pathogenesis of neurodegenerative diseases [32]. Cholesterol transport by astrocytes to neurons is less efficient in HD, with decreased expression of *ApoE* mRNAs, and a lower release of APOE by astrocytes expressing mHTT [21,33]. Since CYP46A1 restores *ApoE* mRNA expression in the striatum of zQ175 mice [17], we quantified its expression in neurons and astrocytes. We focused on nuclear *ApoE* mRNA to quantify neo-synthesized transcripts (Figure 6A,C). *ApoE* mRNA dots quantification did not show difference in MSNs between WT-GFP, HD-GFP and HD-CYP46A1 mice (Figure 6B). However, as expected, *ApoE* dots were significantly lower in the astrocytes of HD-GFP mice as compared to WT-GFP mice (Figure 6D). Moreover, CYP46A1 restored *ApoE* mRNA expression in astrocytes of HD (Figure 6D). 

## 3. Discussion

Cholesterol metabolism dysregulation plays a critical role in HD pathogenesis and recent studies highlight the interest in considering this pathway as a therapeutic target [7,34]. The neuronal-enriched enzyme CYP46A1 is downregulated in HD transgenic and knock-in mouse models and its striatal restoration alleviates HD phenotypes [16,17]. In particular, CYP46A1 reinstates cholesterogenesis, cholesterol efflux and catabolism. We developed a co-labelling method that combined mRNA in situ hybridization and the immunofluorescence detection of MSNs and astrocytes on mouse brain sections to assess CYP46A1-mediated regulation of cholesterol pathway genes in these two cellular populations. We found that CYP46A1 exerts a bimodal effect on cholesterol metabolism, one in neuronal cells (MSNs) on key cholesterogenesis genes, and one in astrocytes, likely via paracrine effects on cholesterol transport/efflux. 

The characterization of mRNA distribution in different cell types such as neurons or astrocytes, is now possible on a single brain slice. NeuN and S100 immunodetection associated with RNAscope^®^ has been developed but as mentioned by other studies [17,31], technical improvements are still necessary. The combination of RNAscope^®^ and immunostaining impairs immunolabelling quality probably due to protease digestion treatment. Indeed, the protease treatment modified the immunolabelling quality by altering antibody binding to antigen. We were required to reduce the protease incubation period and to increase the post-fixation time of the antibody-protein complex to improve immunolabelling signals. Additionally, we were unable to compare the dot quantification of mRNA between both cell types, probably because each immunostaining affects FISH signals differently. RNAscope^®^ provides the advantages of in situ analysis of mRNA with a single-molecule visualization [35], but in case of abundant mRNA (e.g., *ApoE* mRNA), the dots cannot be differentiated and quantification required the use of macros in ImageJ.

In our previous study, we showed that CYP46A1 restoration had a strong impact on cholesterol metabolism by increasing 24S-OHC, the product of cholesterol degradation but also cholesterol precursor levels [17]. In the adult brain, cholesterol is mainly synthesized by astrocytes [7]; however, depending on the brain physiopathogenesis, one cannot exclude that a re-activation of cholesterol synthesis may occur in neurons. Indeed, in co-cultures of neurons and astrocytes, neurons can synthesize cholesterol, but at a lower rate as compared to astrocytes, probably because of the high energy cost of this metabolic pathway [32]. In HD mouse models, transcriptomic studies on sorted neurons and astrocytes highlight a cell-intrinsic pathology across mouse models of HD [36,37,38]. However, results may differ depending on HD mouse models (R6/2 transgenic which express N-terminal part of mHTT versus zQ175 knock-in mice expressing the full length mHTT). For the R6/2 mice-derived-astrocytes, the most altered pathways are related to fatty acid and cholesterol metabolism whereas opioid signaling, calcium signaling, and synaptogenesis are the core altered pathways in neurons [37,38]. In zQ175, astrocytes do not show downregulation of cholesterogenesis genes, which reveals the transcriptomic differences between glia expressing truncated mHTT versus full-length mHTT [37]. Our results are consistent with these studies as none of the three cholesterogenesis key mRNAs tested (*Hmgcr*, *Dhcr24*, and *Srebp2*) are downregulated in zQ175 astrocytes. The downregulation of *ApoE* mRNA that we observed in zQ175 astrocytes corroborates previous results seen in cultures of HD astrocytes [21,33]. 

Our results, combined with our previous studies [16,17] suggest that the neuronal enzyme CYP46A1 is able to reinstate cholesterol metabolism specifically and locally in neurons with an upregulation of *Hmgcr*, *Dhcr24* mRNA in HD mice. This effect could be due to SREBP2, the key transcription factor regulating cholesterogenesis, which is upregulated in HD neurons expressing CYP46A1. We can therefore exclude a possible contribution of astrocytes in the CYP46A1-mediated increase of cholesterol metabolism. However, we propose that 24S-OHC produced by CYP46A1 in neurons may increase *ApoE* mRNA expression in astrocytes through a paracrine effect in HD mice. Indeed, 24S-OHC is a ligand of LXR positively regulating transcription of *ApoE* [22] that encodes cargo proteins for cholesterol transport from astrocytes to neurons [39,40]. Therefore, both cell populations need to be considered in the CYP46A1-mediated beneficial effect in HD mice.

The coupling of FISH with immunostaining can therefore be used to study the distribution of mRNAs in astrocytes and neurons in the HD brains of mice. CYP46A1 increases cholesterogenesis in neurons and cholesterol efflux on astrocytes, probably through a paracrine effect. This study, focused on neurons and astrocytes, could be extended to other cell types involved in HD physiopathology, such as microglia and oligodendrocytes. 

## 4. Materials and Methods

### 4.1. Mice

Four-month-old littermate WT or heterozygous zQ175 mice were used. zQ175 mice (B6J.129S1-Htttm1Mfc/190ChdiJ) were obtained from Jackson Laboratories, Bar Harbor, ME, USA. All mice used in the study were from the first or second offspring, and the genotype was determined by polymerase chain reaction (PCR) using genomic DNA extracted from the tail or ear. Both males and females were housed in groups with a 12-h light/dark cycle, provided with food and water ad libitum, and kept at a constant temperature (19–22 °C) and humidity (40–50%). All experiments performed on animals followed the European Community guidelines (2010/63/EU) and the French Directive for animal experimentation (2013/118) for the use and care of experimental animals and the requirements for the three Rs for Animal Welfare. The ethics committee and the French Ministry of Research (#17424) approved the animal study protocol.

### 4.2. Production and Stereotaxic Injection of AAVrh10.GFP and AAVrh10.CYP46A1.HA

All AdenoAssociatedVirus (AAV) vectors were obtained by Atlantic Gene therapies (Inserm U1089, Nantes, France). The viral constructs for AAVrh10-GFP and AAVrh10-CYP46A1-HA contained the expression cassette consisting of either the GFP or the human CYP46A1, driven by a CMV/β-actin hybrid promoter (CAG) surrounded by inverted terminal repeats of AAVrh10. The stereotaxic coordinates were: 1 mm rostral to the bregma, 2 mm lateral to the midline and 3.25 mm ventral to the skull surface. The rate of injection was 0.2 µL/min with a total volume of 2 µL per striatum (equivalent to 3.10^9^ genomic particles).

### 4.3. Brain Section Preparation

Five months after stereotaxic injections, mice were deeply anesthetized by intraperitoneal injection of euthazol (150 mg/kg). Intracardiac perfusion of 4% paraformaldehyde in 0.1 M Na2HPO4/NaH2PO4 buffer, pH 7.5 was performed, and brains were stored overnight in the same solution at 4 °C. Then, brains were transferred to a cryoprotective solution containing 30% sucrose and store at −20 °C. Coronal brain sections (30 µm) were performed using a cryostat (Leica^®^, Wetzlar, Germany) and sections were stored at −20 °C in glycerol and ethylene glycol phosphate buffer.

### 4.4. Immunostaining

Brain sections were incubated with primary antibodies overnight at 4 °C: mouse anti-NeuN (1:500; Millipore, Merck KGaA, Darmstadt, Germany); and rat anti-HA (hemagglutinin) (1:400; Roche, Basel, Switzerland). Secondary antibodies (anti-mouse Cy3 (1:500; Merck KGaA, Darmstadt, Germany), anti-rat Alexa Fluor 488 (1:500; ThermoFische, Waltham, MA, USA), were incubated in 5% NGS (normal goat serum) in phosphate buffer saline (PBS) for 2 h at room temperature (RT). Sections were then labelled with Hoechst solution to stain nuclei and mounted under coverslips in Prolong Gold Antifade reagent (Invitrogen, Carlsbad, CA, USA).

### 4.5. FISH Coupled with Immunostaining

Transduced regions were systematically visualized by either GFP fluorescence or HA immunostaining (expression of CYP46A1) to select sections for the FISH assay. FISH was performed using specific fluorophore-coupled RNAscope^®^ (Minneapolis, MN, USA) probes against *Hmgcr*, *Dhcr24*, *Srebp2* and *ApoE*. The RNAscope^®^ was coupled with immunochemistry according to the manufacturer’s protocol of RNAscope^®^ Multiplex Fluorescent Reagent Kit v2 (BioTechne, Minneapolis, MN, USA). Brain sections were first incubated in hydrogen peroxide (H_2_O_2_) (BioTechne) for 10 min at RT. The sections were washed in Tris-buffered saline (TBS) with Tween^®^ (50 mM Tris-Cl, pH 7.6; 150 mM NaCl; 0.1% Tween 20) at RT and mounted on Super Frost treated glass slides. Then, sections were dried twice for 1 h at RT (with a quick immersion in deionized water in between), incubated for 1 h at 60 °C in a dry oven, and dried again overnight at RT in the dark. After rapid immersion in deionized water at RT for rehydration, the excess liquid was removed with absorbing paper (repeated at each step) and a hydrophobic barrier was drawn. A rapid immersion in pure ethanol was performed and slides were incubated at 100 °C in a steamer with a drop of RNAscope^®^ Target Retrieval Reagent (Biotechne) for 15 min. After three washes in deionized water at RT, a last wash of TBS-Tween^®^ was performed. Sections were then incubated overnight with primary antibodies: mouse-anti NeuN (1:500; Millipore) in a Co-detection antibody diluent (BioTechne) or rabbit anti-S100β (ready to use; Dako, Santa Clara, CA, USA) at 4 °C. Brain sections were post-fixed with cold PFA-PBS for 30 min at RT, followed by treatment with RNAcope^®^ Protease Plus (BioTechne) for 30 min at 40 °C in a humid box (to unmask the mRNAs). After three washes in deionized water, brain sections were incubated with hybridization probes of interest (2 h at 40 °C in a humid box) followed by amplification (30 and 15 min at 40 °C) and revelation of RNAscope^®^ signals with Opals at different wavelengths: Opal520, Opal 620 and Opal 650 (30 min at 40 °C; 1:1500 to 1:3000 depending on the RNAscope^®^ probes; Akoya Biosciences, Marlborough, MA, USA) with washes between each step. Finally, the last immunofluorescence step was performed by incubating the secondary antibodies with Co-detection antibody diluent (BioTechne) for 30 min at RT: anti-mouse Cy3 (1:500; Merck), anti-rabbit Cy3 (1:500; Merck). Sections were then incubated with DAPI solution to stain nuclei and mounted under coverslips in Mowiol (Sigma Aldrich, Saint-Louis, MO, USA).

### 4.6. Image Acquisition and Analysis

Image stacks were taken using a confocal laser-scanning microscope (SP5, Leica Microsystems, Wetzlar, Germany), with a pinhole aperture set to 1 Airy unit. Stacks of confocal images were done using a ×10 and ×40 objectives (tropism and neuronal transduction analysis) or a ×63 oil objective (RNA detection microscopy analysis), with a 0.3 µm z-interval. Laser intensity and detector gain were constant for all images of the same analysis. The number of nuclear dots in neurons and astrocytes was quantified in the AAV-transduced sections using imaging tools described in the result session (Figure 2). Then, the average number of RNAscope^®^ dots per cell was quantified by dividing the total number of dots obtained in all selected cells by the number of neurons or astrocytes analyzed. For each condition, around 400 neurons and 100 astrocytes were analyzed. For each analysis, 3 to 4 mice were used (3 for the WT-GFP group, 4 for the HD-GFP group and 4 for the HD-CYP46A1 group) for a total of 400 neurons and 100 astrocytes analyzed in each condition.

### 4.7. Statistical Analysis

Statistical analysis was performed with GraphPad Prism 6 software. All data are represented as mean ± SEM. Statistical significance for RNA dots quantification was evaluated using a one-way ANOVA followed by Kruskal–Wallis post hoc test.

## Figures and Tables

**Figure 1 ijms-24-11001-f001:**
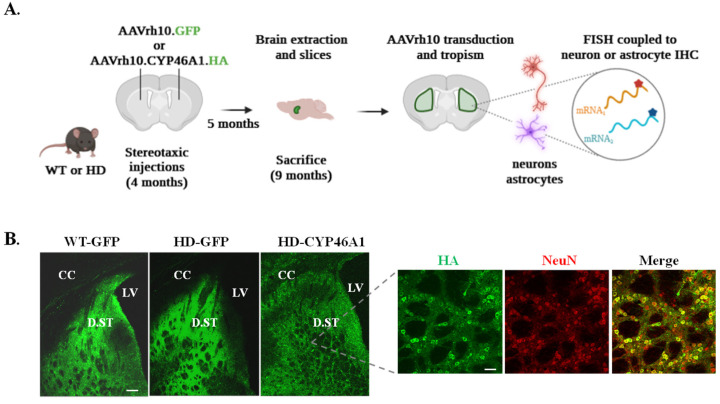
Validation of striatal expression of CYP46A1 in mice: (**A**) experimental set up to assess mRNA quantification on brain sections within neurons and astrocytes. Created with BioRender.com, accessed on 1 January 2022; and (**B**) left and middle panels: expression of GFP in WT (WT-GFP) and HD (HD-GFP) mice after stereotaxic injection of AAV-GFP. Right panels: expression of CYP46A1-HA revealed by HA immunostaining in HD (HD-CYP46A1) mice after stereotaxic injection of AAV-CYP46A1-HA on striatal sections (Scale bar: 100 µm). A close-up of CYP46A1-HA in HD is shown after double HA/NeuN immunostaining (scale bar: 30 µm) and reveals neuronal tropism of AAVrh10. CC: corpus callosum, LV: Lateral Ventricle, D.ST: Dorsal Striatum.

**Figure 2 ijms-24-11001-f002:**
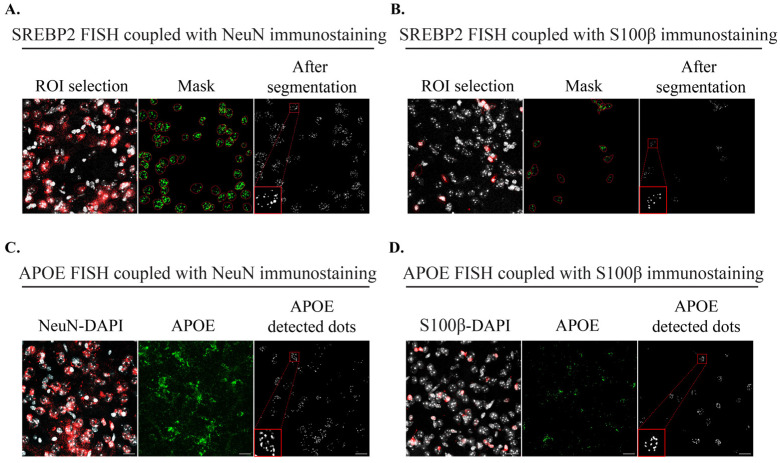
Imaging methods after FISH coupled with IHC: (**A**) sequence of the different stages for *Srebp2* signal quantification in neurons (NeuN immunostaining in red); and (**B**) in astrocytes (S100β immunostaining in red): contouring of nucleus labeled with DAPI (in white) from positive immunostained cells, mask application to the *Srebp2* signal and segmentation result; (**C**) stages of macro for *ApoE* signal quantification in neurons (NeuN immunostaining in red); and (**D**) astrocytes (S100β immunostaining in red): NeuN/S100β-DAPI labelling, *ApoE* signal before macro treatment, mask and Gaussian result (after macro). Scale bar: 17 µm.

**Figure 3 ijms-24-11001-f003:**
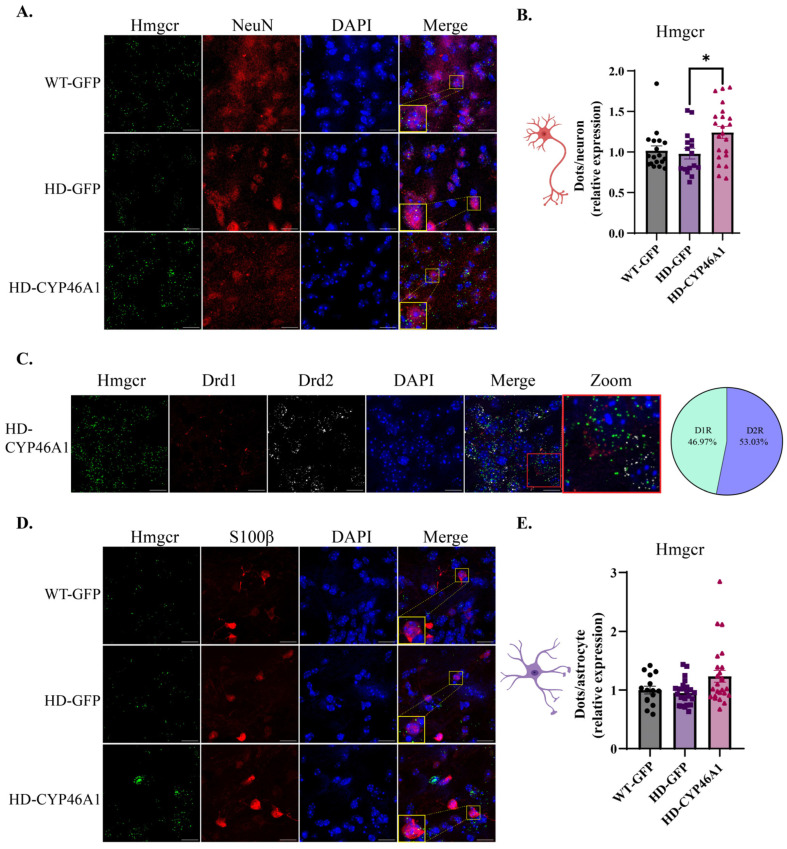
*Hmgcr* gene expression in neurons and astrocytes after CYP46A1-HA expression in HD mice: (**A**) mice were injected with AAVrh10-GFP (WT GFP and HD GFP) or AAVrh10-CYP46A1-HA (HD-CYP46A1). RNAscope^®^ coupled with NeuN immunostaining was performed on WT and HD brain mouse sections and images were taken at 63× objective (Scale bar = 17 μm); (**B**) quantification of dots on approximately 400 neurons * *p* < 0.05 (HD-GFP vs. HD-CYP46A1); (**C**) RNAscope^®^ discrimination between neurons expressing *Drd1* and *Drd2* in the HD-CYP46A1 group and proportion of *Hmgcr* in each D1 or D2 MSNs in this same group; (**D**) RNAscope^®^ coupled with S100β immunostaining (Scale bar = 17 μm); and (**E**) quantification of dots on approximately 100 astrocytes. Results are expressed as mean ± SEM (n = 3–4). One-way ANOVA with Kruskal–Wallis post hoc test was used for statistical analysis.

**Figure 4 ijms-24-11001-f004:**
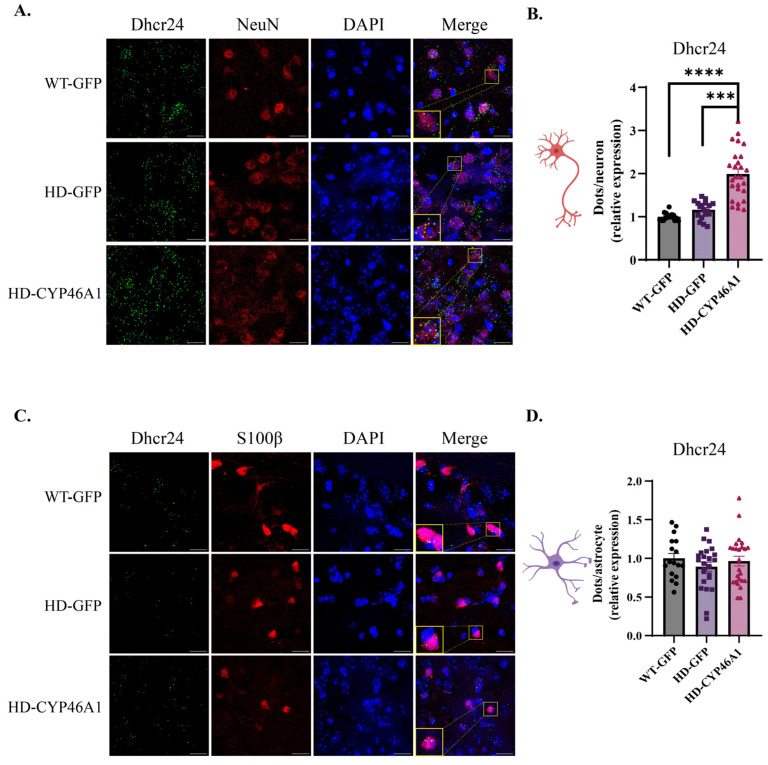
*Dhcr24* gene expression in neurons and astrocytes after CYP46A1-HA expression in HD mice: (**A**) mice were injected with AAVrh10-GFP (WT-GFP and HD-GFP) or AAVrh10-CYP46A1-HA (HD-CYP46A1). RNAscope^®^ coupled with NeuN immunostaining was performed on WT and HD brain mouse sections and images were taken at 63× objective (Scale bar = 17 μm); (**B**) quantification of dots on approximately 400 neurons **** *p* < 0.0001 (WT-GFP vs. HD-CYP46A1) and *** *p* < 0.001 (HD-GFP vs. HD-CYP46A1); (**C**) RNAscope^®^ coupled with S100β immunostaining (Scale bar = 17 μm); and (**D**) quantification of dots on approximately 100 astrocytes. Results are expressed as mean ± SEM (n = 3–4). One-way ANOVA with Kruskal–Wallis post hoc test was used for statistical analysis.

**Figure 5 ijms-24-11001-f005:**
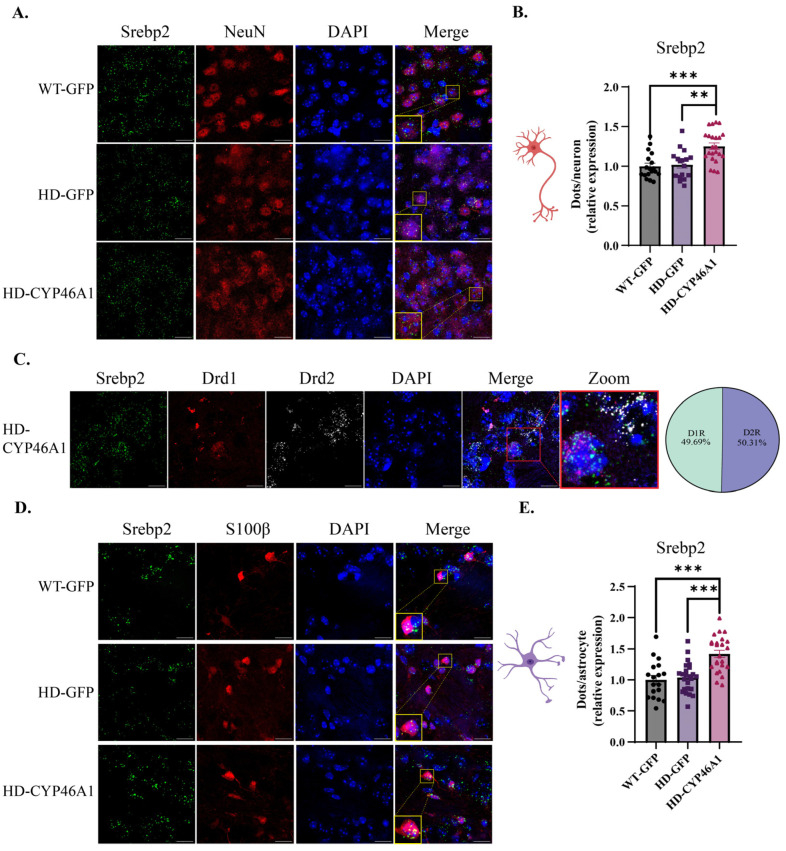
*Srebp2* gene expression in neurons and astrocytes after CYP46A1-HA expression in HD mice: (**A**) mice were injected with AAVrh10-GFP (WT-GFP and HD-GFP) or AAVrh10-CYP46A1-HA (HD-CYP46A1). RNAscope^®^ coupled with NeuN immunostaining was performed on WT and HD brain mouse sections and images were taken at 63× objective (Scale bar = 17 μm); (**B**) the number of dots in neurons was quantified on approximately 400 neurons *** *p* < 0.001 (WT-GFP vs. HD-CYP46A1) and ** *p* < 0.005 (HD-GFP vs. HD-CYP46A1); (**C**) RNAscope^®^ discrimination between neurons expressing *Drd1* or *Drd2* in HD-CYP46A1 group and proportion of *Srebp2* in each D1 or D2 MSN in this same group; (**D**) RNAscope^®^ coupled with S100β immunostaining (Scale bar = 17 μm). (**E**) quantification of dots on approximately 100 astrocytes. Results are expressed as mean ± SEM (n = 3–4). One-way ANOVA with Kruskal–Wallis post hoc test was used for statistical analysis.

**Figure 6 ijms-24-11001-f006:**
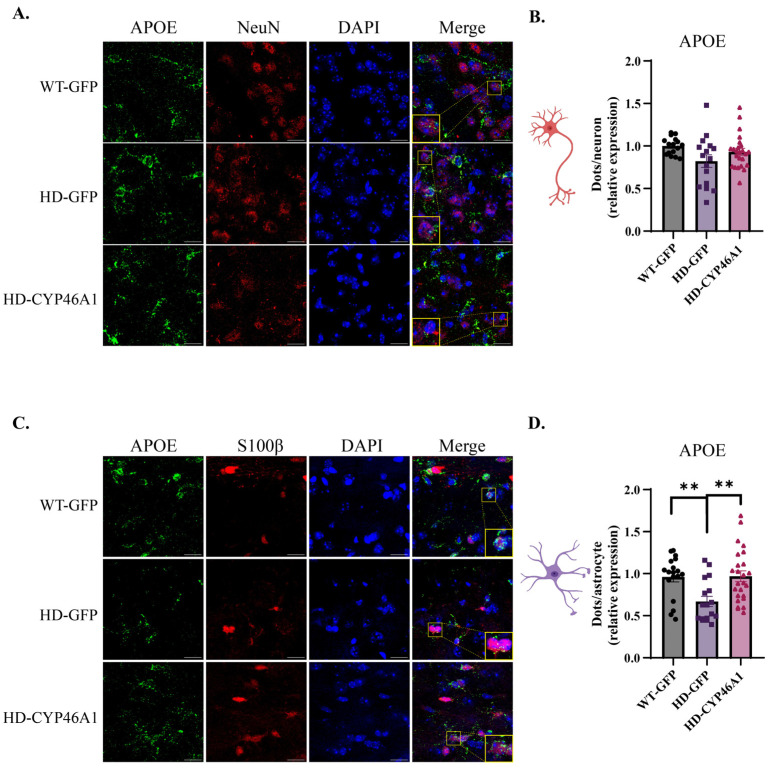
*ApoE* gene expression in neurons and astrocytes after CYP46A1-HA expression in HD mice: (**A**) mice were injected with AAVrh10-GFP (WT-GFP and HD-GFP) or AAVrh10-CYP46A1-HA (HD-CYP46A1). RNAscope^®^ coupled with NeuN immunostaining was performed on WT and HD brain mouse sections and images were taken at 63× objective (Scale bar = 17 μm); (**B**) quantification of dots on approximately 400 neurons; (**C**) RNAscope^®^ coupled with S100β immunostaining (Scale bar = 17 μm); and (**D**) Quantification of dots on approximately 100 astrocytes ** *p* < 0.05 (WT-GFP vs. HD-GFP and HD-GFP vs. HD-CYP46A1). Results are expressed as mean ± SEM (n = 3–4). One-way ANOVA with Kruskal–Wallis post hoc test was used for statistical analysis.

## Data Availability

Data available upon request from the authors.

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
