# Peer review of "Cell-Type Specific Regulation of Cholesterogenesis by CYP46A1 Re-Expression in zQ175 HD Mouse Striatum"

_ijms, 2023, doi:10.3390/ijms241311001_

Round 1
Reviewer 1 Report
This is a study of cholesterol metabolism in the zQ175 mouse model of Huntington's Disease (HD). The authors combined adeno-associated virus (AAV)-mediated expression of cholesterol-24-hydroxylase protein (CYP46A1) with a hemagglutinin (HA) tag (or green fluorescent protein (GFP) control) with immunofluorescent localization of NeuN or s100 proteins in striatum as markers for neurons or astrocytes, respectively. They also used fluorescent in situ hybridization (FISH) to localize mRNA's for several cholesterol metabolizing genes in the zQ175 mice stereotactically injected to increase CYP46A1 expression.
This is an elegant study in terms of techniques, but it has several shortcomings that the authors can readily correct. First, there is no mention of mouse survival. They examined mice 5 months after stereotactic injection. Was there comparable mortality among all groups, or did the zQ175 mice survive less? (If this was the case, there may be some selection bias) Did CYP46A1 expression alter survivability? Second, the authors note that the FISH technology they use involved a protease digestion step. This theoretically could lessen NeuN and/or S100 proteins that they immunostained for to detect striatal neurons or astrocytes. Did this occur, and did it in any way skew their results? Did they control for this possibility? Third, they did not control for the CYP46A1 expression, as they only injected the AAV vector for this protein only into HD model mice (zQ175) and not WT controls. When they examined gene expression of other cholesterol genes, they compared the WT mice with GFP injections to the HD mice that had CYP46A1 injections. Without having WT mice also receive the CYP46A1 vector, one really doesn't know if increased expression of CYP46A1 has anything to do with the HD model used.
These problems can be addressed verbally in the Discussion, and additional experiments are probably not needed (I HATE "reviewer experiments"). However, these deficits detract from what is otherwise an elegant study.
Author Response
Dear Reviewer 1,
We really want to thank reviewer 1 for the positive feedback.
- Point 1 : Regarding the first comment on HD mouse survival : the zQ175 mouse model do not display any decrease of lifespan. Moreover, in our previous manuscripts (Boussicault et al, 2016 and Kacher et al, 2019), we showed that CYP46A1 has benefic effects on HD phenotype without any toxicity described during the time of the studies. We had this comment on the result part (Line 102-104)
- Point 2 : This is a very good point which needs clarifications. Immunolabelling is necessary to identify cell population is not quantified per se. DAPI staining is used to select ROI in which FISH signals are quantified. Therefore, the quality of immunolabelling cannot alter the FISH quantifications. We adress this point in the discussion part (lines 280-281 and lines 284-286)
- Point 3 : this is again a really good point. The aim of the study was to adress CYP46A1 effect on HD cells but not on WT cells. Therefore, we did not transduced WT cells with AAVrh10-CYP46A1.HA. However, we do think that we may find the same effect of CYP46A1 on gene expression in WT cells. Personal data showed that WT-CYP46A1 mice had increase of spine density as compared to WT; this effect was also observed in HD-CYP46A1 mice as compared to HD mice(results published KAcher et al, 2019) . In our manuscript we wanted to compare HD-CYP46A1 mice with a physiologic phenotype : WT mice. This was important for rescue results, especially for ApoE FISH data.
Reviewer 2 Report
The manuscript ijms-2470233 entitled Cell-type specific regulation of cholesterogenesis by CYP46A1 re-expression in zQ175 HD mouse striatum by K. Pinchaud and coworkers investigates the effect of restoration of the neuronal cholesterol 24-hydroxylase, CYP46A1, because its expression in striatal neurons of HD mouse models is beneficial for motor behavior, cholesterol metabolism, transcriptomic activity and alleviates neuropathological hallmarks induced by mHTT.
They quantified the distribution of key genes regulating cholesterol metabolism and efflux in astrocytes and neurons using in situ hybridization coupled with S100beta and NeuN immunostaining. Neuronal expression of CYP46A1 in the striatum of HD zQ175 mice increased key cholesterol synthesis driver genes (Hmgcr, Dhcr24) specifically in neurons. This effect was associated with an increase of the srebp2 transcription factor gene that regulates most of the genes encoding for cholesterol enzymes. However, the cholesterol efflux gene, ApoE, was specifically upregulated in astrocytes by CYP46A1 probably though a paracrine effect. The results suggest a bimodal effect of CYP46A1: one in MSNs to reinstate cholesterol biosynthesis and one in astrocytes to favor cholesterol efflux. The beneficial effect of CYP46A1 in HD reflects the dynamic balance between cholesterol synthesis, uptake, and export.
The research aim is clear.
Experiments are appropriate and significative, and the methodology used robust.
Figures are clear and informative.
Discussion is consistent with the results.
English language is rather good.
Minor comments
Line 92: the title should be together with the following text
Line 230: footnotes to the figure 5 are divided by two pages.
Line 406: results should be presented as mean ± SD. The use of SEM is inappropriate since it is not inferential statistic.
English lenguage is good.
Author Response
Dear reviewer,
We want to thank reviewer 2 for the very positive feedback to our manuscript.
- We modified the location of the title (Line93)
- We introduced the figures and footnotes insinde the manuscript to falicitate reviewer reading. We would let IJMS editors adjusting the correct location of figures and footnotes in the final manuscript.
- We thank the reviewer for the comment related to statistics. We are aware that we may have done some confusions between data description and generalization about a population in the discussion part. All along the manuscript, we describe the characteristics of our data set and we summarized data without any prediction. As a consequence, we made precisions in the discussion part to avoid the generalization of our data set (see lines 275, 313, 315, 320 and 325 : conclusions are restricted to HD mice). As a result, we propose to keep the descriptive statistics with mean± SEM.
Round 2
Reviewer 1 Report
In their response the authors have addressed my initial concerns. As the authors realize, combining the FISH technique with immunolabeling is problematic. In any event, that the increased expression of CYP46A1 following stereotactic striatal injection of an AAV vector ameliorates the HD phenotype is, of course, the primary observation. The present paper further characterizes changes in cholesterol metabolism in striatal neurons and astrocytes brought about by that injection. That is a valuable mechanistic contribution.